# Measuring and Improving Robustness of Deep Neural Networks

## Abstract

Deep neural networks perform well on train data, but are often unable to adapt to data distribution shifts. These are data which are rarely encountered, and thus are under-represented in our training data. Examples of this includes data under adverse weather conditions, and data which have been augmented with adversarial perturbations. Estimating the robustness of models to data distribution shifts is important in enabling us to deploy them into safety critical applications with greater assurance. Thus, we desire a measure which can be used to estimate robustness. We define robustness in 4 ways: Generalization Gap, Test Accuracy (Clean & Corrupted), and Attack Success Rate. A measure is said to be representative of robustness when consistent (non-contradicting) relationships are found across all 4 robustness definitions. Through our empirical studies, we show that it is difficult to measure robustness comprehensively across all definitions of robustness, as the measure often behave inconsistently. While they can capture one aspect of robustness, they often fail to do so in another aspect. Thus, we recommend that different measures be used for different robustness definitions. Besides this, we also further investigate the link between sharpness and robustness. We found that while sharpness has some impact on robustness, this relationship is largely affected by the choice of hyperparameters such as batch size.

## 1 Introduction

Deep Neural Networks (DNNs) provide state-of-the-art performances across various visual tasks. However, a key problem exists when deploying DNNs in real-world conditions. These DNNs often encounter out-of-distribution (OOD) data during deployment. This can come in the form of differing environmental conditions (e.g., rain, haze, fog). However, perhaps most concerning of all are adversarial attacks (Szegedy et al., 2013; Goodfellow et al., 2014). Adversarial attacks aim to fool DNNs into making incorrect decisions. Given the critical use cases of DNNs in our applications, there is a need to both measure and improve the robustness of DNNs to OOD data in the wild. Doing so would provide us with assurances that our DNNs are safe and robust when deploying them in the real-world. In this work, we focus on identifying a measure of robustness for DNNs. This measure will serve as a metric to determine how robust an arbitrary DNN is. In our experiments, we study the existence of such a measure for the Image Classification task. To quantify robustness, we use the Generalization Gap, Clean Test Accuracy, Corruption Test Accuracy, and the Attack Success Rate (ASR). A measure is said to be representative of robustness if it consistently achieves a high correlation across all these definitions of robustness. In our experiments, we adopt the approach taken by Jiang et al. (2019); Dziugaite et al. (2020), which conducted large scale empirical studies to discover correlations between their introduced measures and the Generalization Gap. However, in addition to the measures introduced by Jiang et al. (2019); Dziugaite et al. (2020) we use other measures such as boundary thickness (Yang et al., 2020) and gradient norm measures (Ross & Doshi-Velez, 2018). Furthermore, we extend this study to consider the relationship between the measures and Test Accuracy (Clean & Corrupted images). We also study their relationship with the ASR of various adversarial attacks. We then follow up this study by analyzing the significance of each measure across the different definitions of robustness. Our experiments show that none of the measures we studied are consistent across all definitions of robustness. Conflicting relationships with the different definitions of robustness are formed. Additionally, we found that the choice of hyperparameters such as batch size significantly influences the robustness of DNNs. This calls the reliability of these

measures into question. These findings lead us to conclude that there is no one measure that can comprehensively reflect the robustness of DNNs. Thus, we recommend that separate measures for the different robustness definitions be used. As we are concerned with OOD data, we focus our recommendations on measures for Corruption Test Accuracy and ASR. When concerned with Corruption Test Accuracy, we found the *weight gradient norm* and *hessian eigenvalue* (sharpness) to best reflect it. On the other hand, when concerned with ASR, we found *boundary thickness* to be most representative of it. Besides identifying a measure for robustness, we also investigated the relationship between sharpness and robustness. While substantial number of works have advocated that flatness of loss landscape leads to improved robustness (Foret et al., 2020; Kwon et al., 2021), other works (Dinh et al., 2017; Andriushchenko et al., 2023) have proven otherwise. The discrepancies in these studies lead us to conduct this investigation. We found that while low sharpness can lead to improved robustness, this relationship is significantly influenced by the choice of batch size used when training DNNs.

We summarize our contributions and findings below:

- Conducted large scale empirical studies to find a measure for robustness. Different from previous studies, we capture robustness more comprehensively by considering it from 4 different angles. In terms of the Generalization Gap, Test Accuracy (Clean & Corrupted), and ASR.

- Identified *hessian eigenvalue* and *weight gradient norm* to be most promising when concerned with Corruption Test Accuracy. Additionally, we found *boundary thickness* to be the most promising measure of robustness when concerned with ASR.

- Demonstrated that the link between sharpness and robustness is significantly impacted by the choice of hyperparameters such as batch size.

## 2 RELATED WORK

While DNNs yield excellent performances on In-distribution (ID) data, they tend to suffer a performance drop when they encounter OOD data. This problem is further exacerbated when they are faced with adversarial examples. Ideally, we want robust DNNs which can both maintain performance on OOD data and are robust to adversarial examples. To improve the robustness of DNNs, we first require a way to measure robustness. This is obviously not as simple as directly measuring the Test Accuracy (Clean & Corrupted) or ASR, as this set of OOD data is generally unknown. What we instead seek is a metric that captures a property of a DNN which is in turn reflective of the DNNs robustness. Most works (Jiang et al., 2019; Dziugaite et al., 2020; Kim et al., 2024) in this field investigate this matter through large scale empirical studies. They train numerous DNNs, perform the relevant measures, before performing correlation analysis with robustness. The measure that yields the highest correlation will be deemed as the most reflective measure of robustness. However, these works are limited in scope. Jiang et al. (2019) only studied the relationship between their selected measures and the Generalization Gap. Andriushchenko et al. (2023) looked into the relationship between both Clean Test Accuracy and the Generalization Gap. However, they neglected the relationship between Corruption Test Accuracy and the ASR. In this work, we argue that it is equally important to consider all aspects of robustness when finding a measure that reflects robustness. Hence, we define the robustness of DNNs in 4 ways. We use the Generalization Gap, Clean Test Accuracy, Corruption Test Accuracy, and ASR to represent robustness. We then utilize measures to perform correlation analysis via the Kendall rank correlation coefficient against all these definitions of robustness. A measure that yields high correlation scores against all robustness definitions will be taken as a reflective measure of robustness. Another question we want to tackle is the relationship between sharpness and robustness. While Jiang et al. (2019) found that their sharpness measures were strongly correlated with robustness (Generalization Gap), Andriushchenko et al. (2023) observed weak correlation between sharpness and robustness. In fact, it is training parameters like the learning rate that influences whether the relationship with robustness is positively or negatively correlated. The contention between these findings leads us to further investigate this matter.

## 3 BACKGROUND

### 3.1 DEFINITIONS OF ROBUSTNESS

**Generalization Gap.** Measures the difference in performance during train and test time. It can be defined as such $Generalization\ Gap = Test\ Error - Train\ Error$. A large Generalization Gap indicates that the DNN performs well on train data (low train error) but does poorly on test data (high test error), indicating poor robustness of a DNN. Hence, we desire a tight Generalization Gap, where test error does not deviate much from train error.

**Clean Test Accuracy.** Measures how well the DNN performs on the test dataset. It can be defined as such $Clean\ Test\ Accuracy = \frac{1(f(x_i),t_i)}{|D_{test}|} * 100\%$, $(x_i, t_i) \in D_{test}$, where $D_{test}$ represents the test dataset, $(x_i,t_i)$ an input-target label pair, and $f$ a trained DNN. The higher the test accuracy, the more robust a DNN is.

**Corruption Test Accuracy.** Measures how well the DNN performs on a corrupted version of the test dataset, and can be defined as such $Corrupted\ Test\ Accuracy = \frac{1(f(x_i^{corr}),t_i)}{|D_{test}|} * 100\%$, $(x_i^{corr}, t_i) \in D_{test}^{corr}$, where $D_{test}^{corr}$ represents the corrupted test dataset, and $x_i^{corr}$ is a data instance from the corrupted test dataset. Corrupted data can be seen as a representation of OOD data. The higher the Corruption Test Accuracy is, the more robust a DNN is.

**Attack Success Rate.** Indicates how effective an adversarial attack is. It measures the proportion of adversarial examples in the test dataset that successfully causes a model to make incorrect predictions. It can be defined as such $Attack\ Success\ Rate = \frac{1(f(x_i^{adv}),t_i^{adv})}{|D_{test}^{adv}|} * 100\%$, $(x_i^{adv}, t_i^{adv}) \in D_{test}^{adv}$, where $D_{test}^{adv}$ represent the set of adversarial examples crafted from the test dataset. $x_i^{adv}$ represents an instance of an adversarial example, with $t_i^{adv}$ being the target corresponding to it. As we are interested in the robustness of DNNs, we want the ASR to be as low as possible. A low ASR indicates that the DNN is robust to adversarial attacks.

### 3.2 MEASURING ROBUSTNESS OF DNNS

We seek a measure that reflects how robust an arbitrary DNN is. This means that given a DNN, by performing this measurement on the DNN, we can use the measurement obtained to estimate the robustness of the DNN. To do so, we first need to identify what we want to measure. Intuitively, these measures should capture the properties of the DNNs. In this subsection, we take a closer look into the measures we used when measuring properties of DNNs. Given the numerous measures we use, we categorised them into 4 categories.

**Complexity Measures.** Complexity-based measures are typically calculated using the weight matrix of trained DNNs. They give us an indication of how complex the learnt function is. Typically, the less complex a solution is, the more generalizable and thus robust the DNN is. In our experiments, we utilize several complexity-based measures based on the norm of the weight matrix. This includes the *number of parameters, L2 norm, Path-norm* (Neyshabur et al., 2015), *Spectral norm*, and *Frobenius norm*. For norm-based complexity measures, smaller measures indicate less complex DNNs. Besides norm-based complexity measures, we also use the *sparsity* of the weight matrix (Liu et al., 2022) as a measure.

**Decision Boundary Measures.** Decision boundary-based measures estimates the distance between class boundaries. A small distance between class boundaries implies that just a small amount of perturbation is required to cross over the class boundaries. This indicates poor robustness. In this work, we consider two measures to estimate decision boundaries. *Inverse margin* and *boundary thickness* (Yang et al., 2020).

**Sharpness Measures.** Sharpness has been linked to robustness (generalizability). The intuition behind this is that with smoother loss landscapes (low sharpness), DNNs would be less sensitive to perturbations. This implies improved robustness. Despite several works supporting this claim, other works have instead found that there is little to no correlation between sharpness-based measures and robustness. Given this conflict, we found it fit to conduct our own study. In our experiments, we consider the *Hessian eigenvalue*, *Hessian trace*, and *Average sharpness* (Andriushchenko et al., 2023) as estimates for the sharpness of DNNs.

**Gradient Measures.** In this work, we study the use of *input gradient norm* (Ross & Doshi-Velez, 2018) and *weight gradient norm* (Zhao et al., 2022) as measures for robustness. These measures have been incorporated as terms to be regularized in the DNN training process. As such, it is not uncommon to associate low gradient norm values with better robustness.

### 3.3 CORRUPTIONS AND ADVERSARIAL ATTACKS

We are interested to understand the DNNs performance on OOD data in the form of corruptions and adversarial examples. In our experiments, we consider 14 common corruptions (Hendrycks & Dietterich, 2019) to represent both noise and adverse weather conditions. For adversarial attacks, we consider only whitebox attacks. Adversarial examples were crafted using the Fast-Gradient-Sign-Method (FGSM) (Goodfellow et al., 2014) and Projected Gradient Descent (PGD) (Madry et al., 2017) algorithms. We chose these algorithms as they provide the best balance between high ASR and compute efficiency.

### 3.4 SHARPNESS OPTIMIZERS

Sharpness optimizers introduces an additional term into the learning objective termed as loss sharpness, which aims to encourage smoother loss landscapes. For this to occur, the optimizers first finds the loss value in the worst case by perturbing the learnt parameters at the current timestep. Thereafter, they minimise this value. This transforms the learning objective into a min-max optimization problem. Through introducing this learning paradigm, they hope to find parameters that lie in flat neighbourhoods (smooth loss landscape) having uniformly low loss. This leads to DNNs which are more robust. By introducing sharpness optimizers into the training pipeline, we hope to obtain DNNs with vastly different loss landscapes and sharpness values. Doing so will help us to perform a more thorough study into the connection between sharpness and robustness.

## 4 EXPERIMENTS

### 4.1 IMAGE CLASSIFIERS

In this work, we look to discover a measure that is reflective of robustness across all 4 definitions for the Image classification task. To perform a comprehensive study to seek convincing measures of robustness, we took an empirical approach. This involves training a large pool of well-trained classifiers with vastly different robustness behaviors. In our experiments, we utilize the Residual Neural Network (ResNet) architecture (He et al., 2016) for our image classifiers. We trained multiple ResNet classifiers under different hyperparameter configurations on the *Imagenette*[1] dataset, training till convergence (cross-entropy 0.01), and repeating each experiment 3 times with different initialization values. Performing this resulted in 486 different hyperparameter configurations and a total of 1458 classifiers. We detail the different hyperparameter configurations in appendix E.1.

### 4.2 GENERATING OUT-OF-DISTRIBUTION DATA

We want to measure our classifiers robustness (performance) when it encounters OOD data. To generate data that is representative of OOD data, we employed various techniques to augment our test dataset.

**Common Corruptions.** We measure the robustness of our classifiers to OOD data in the form of common corruptions. We obtain this corrupted data by running 14 natural perturbations (Hendrycks & Dietterich, 2019) on the test dataset. This includes the addition of noise (*gaussian noise, shot noise, etc...*) and adverse weather conditions (*Frost, Fog, etc...*). Our initial analysis found that majority of the corruptions have little impact on the Test Accuracy, with only *Fog* and *Contrast* causing significant drops in Test Accuracy.

**Adversarial Attacks.** We also measure the robustness of our classifiers to OOD data in the form of adversarial examples. To generate adversarial examples, we use the FGSM and PGD algo-

---

[1]https://github.com/fastai/imagenette

rithms using different attack budget settings. For both algorithms, we ran attacks with budgets $\{2/255, 5/255, 8/255\}$ and calculated their respective ASR.

### 4.3 SHARPNESS OPTIMIZERS

In our experiments, we also want to further understand the relationship between sharpness and robustness. To do so, we trained ResNet classifiers both with and without sharpness optimizers. We utilize 2 variants of sharpness optimizers: Sharpness-Aware Minimization (SAM) (Foret et al., 2020) and Adaptive Sharpness-Aware Minimization (ASAM) (Kwon et al., 2021). Both these methods optimize towards obtaining a local minimum in a smooth region. However, while SAM calculates the worst case via a fixed radius, ASAM is scale invariant and calculates this adaptively. This removes the drawback that SAM has to sensitivity of parameter re-scaling. We hope that by introducing different sharpness optimizers, we can capture more varying properties and behaviors.

### 4.4 MEASURES

To discover a representative measure of robustness, we select various measures, implement them, and measured our 1458 trained classifiers. When performing the measures, there exists hyperparameters to be set. We detail these in appendix E.2. As we have 3 classifiers corresponding to each hyperparameter configuration (just with different seed values), we took the average of the measured values across the 3 classifiers. This results in our subsequent analysis being conducted on 486 classifiers. We hope that by doing so, we can reduce the impact which randomness may have and further increase the validity of our experiments. Following this measurement phase, we perform correlation analysis via the Kendall Rank Correlation Coefficient for each of the measures against the Generalization Gap, Test Accuracy (Clean & Corrupted), and ASR.

## 5 RESULTS AND ANALYSIS

In this section, we present our analysis on the relationships observed between the measures and the different definitions of robustness. We also indicate which measures are most representative of robustness. We particularly do so for measures which reflect robustness in terms of Corruption Test Accuracy and ASR, as we are interested in the case of OOD data. A representative measure is one that behaves consistently and achieves a high correlation score across all robustness definitions.

### 5.1 CORRUPTION TEST ACCURACY DISPLAYS WEAKER CORRELATION COMPARED TO CLEAN TEST ACCURACY

As seen from Figure 1 - 3, across the 4 categories of measures, a common observation is that the correlation of Corruption Test Accuracy tends to be almost half as weak compared to Clean Test Accuracy. This phenomenon holds true for each measure within each category. We attribute the drop in correlation for Corruption Test Accuracy to the random perturbations introduced during the corruption process.

### 5.2 INCONSISTENCIES OCCURS ACROSS THE DIFFERENT ROBUSTNESS DEFINITIONS

Another observation made is that inconsistencies arises from our correlation analysis. While a measure might obtain high correlation scores with multiple robustness definitions, the correlation obtained (positive or negative) might have different implications on robustness. These implications are sometimes counter-intuitive to one other, bringing the effectiveness of these measures into question. We describe such instances in greater detail in the following subsection.

### 5.3 CORRELATION ANALYSIS BY THE CATEGORIES

**Complexity-based Measures.** As seen in Figure 1, aside from *sparsity*, all other measures had a correlation score $< |0.2|$ for all robustness definitions. This indicates that they are not indicative measures of robustness. Taking a closer look into *sparsity*, despite *sparsity* obtaining a correlation score of $> |0.2|$ for both Generalization Gap and Test Accuracy (Clean & Corrupted), these results

are contradictory to each other. A positive correlation with Test Accuracy means that high *sparsity* (low complexity) yields higher Test Accuracy (improved robustness). While this is desirable, the positive correlation with Generalization Gap means that high *sparsity* (less complex) leads to a higher Generalization Gap (weaker robustness). The contradictory result calls the reliability of *sparsity* as a measure of robustness into question. Furthermore, the correlation for *sparsity* against the ASR is weak.

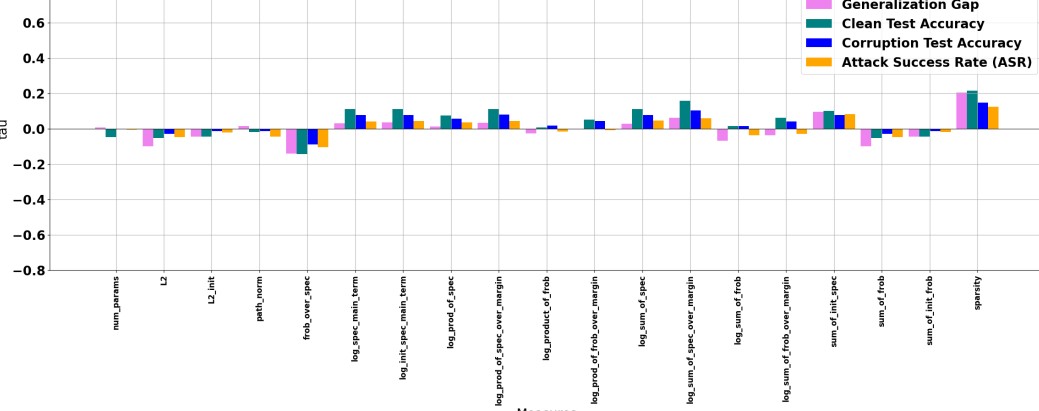

Figure 1: Correlation scores when correlating complexity-based measures against the different definitions of robustness.

**Decision boundary-based Measures.** As seen in Figure 2, all 3 measures in this category scored a correlation score of $> |0.2|$ when correlated with Generalization Gap, and Test Accuracy (Clean & Corrupted). These measures also scored just below $|0.2|$ when correlated with ASR. However, as with the case of complexity-based measures, inconsistencies arise.

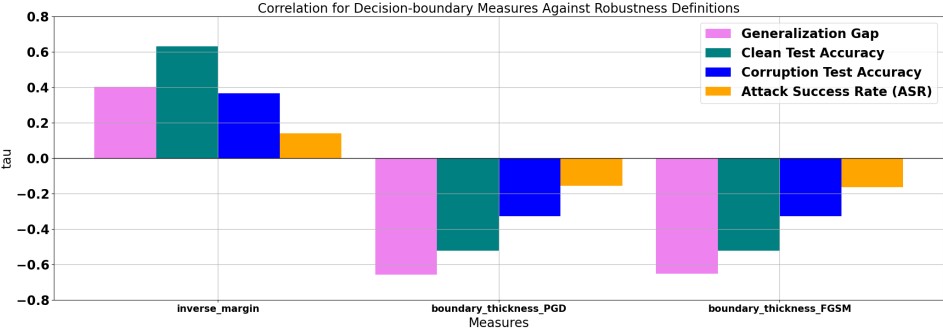

Figure 2: Correlation scores when correlating decision boundary-based measures against the different definitions of robustness.

- **Inverse margin.** Positive correlations were obtained for all robustness definitions. This means large margins lead to lower Generalization Gap (improved robustness). Positive correlation with ASR means large margins lead to lower ASR (improved robustness). However, a positive correlation with both Clean and Corruption Test Accuracy means larger margins leads to lower Test Accuracy (weaker robustness). This highlights the inconsistencies that arise from what seemed to be promising measures.

- **Boundary thickness.** Calculated with respect to both FGSM and PGD, negative correlations were obtained across all robustness definitions. This means that thicker boundary leads to both lower Generalization Gap and lower ASR (improved robustness). However, our findings also implied that thicker boundaries lead to lower Test Accuracy (weaker robustness). This again emphasises the inconsistencies.

**Sharpness-based Measures.** As seen in Figure 3, all sharpness-based measures displayed similar trends, they obtained negative correlation with all robustness definitions. Across all sharpness-based measures, only Test Accuracy consistently obtained a correlation score $> |0.2|$. On the other hand, the correlation with ASR and Generalization Gap was particularly weak across most sharpness measures. This is apart from *hessian eigenvalue* which displays the strongest relationship with respect to all robustness definitions among the sharpness-based measures. In particular, we note that when correlated with Generalization Gap, it obtained a score close to -0.2. Seeing as how *hessian eigenvalue* appears as the most significant sharpness-based measure, we focus our discussion on it. The negative correlation with Test Accuracy implies that lower sharpness (smoother loss landscape) leads to higher Test Accuracy (improved robustness). This is consistent with works that prove that smooth minima lead to improvements in robustness. However, the negative correlation score (-0.2) of *hessian eigenvalue* with Generalization Gap implies that low sharpness leads to higher Generalization Gap (weaker robustness). This observation reiterates the inconsistencies.

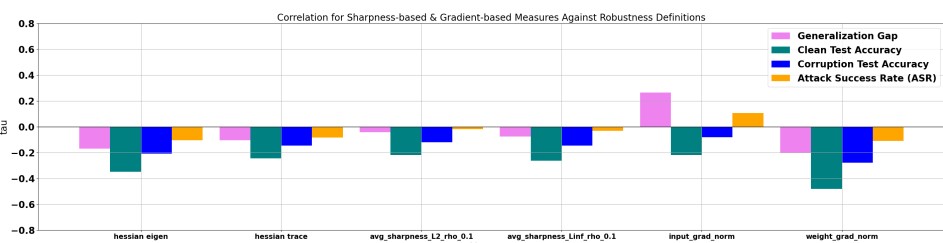

Figure 3: Correlation scores when correlating both sharpness-based and gradient-based measures against the different definitions of robustness.

**Gradient-based Measures.** As seen in Figure 3, both gradient-based measures display weak correlation with ASR. This indicates their inability to capture the relationship with ASR.

- **Input gradient norm.** A positive correlation is obtained when correlated with Generalization Gap. This implies that a lower *input gradient norm* leads to a lower Generalization Gap (improved robustness). The negative correlation with Clean Test Accuracy means lower *input gradient norm* leads to higher Clean Test Accuracy (improved robustness). In this regard, *input gradient norm* is consistent across these 2 robustness definitions. However, it is unable to capture the relationship for Corruption Test Accuracy and ASR.

- **Weight gradient norm.** A negative correlation is obtained across all 4 robustness definitions. Negative correlation for Generalization Gap implies that lower *weight gradient norm* leads to higher Generalization Gap (weaker robustness). On the other hand, negative correlation with Test accuracy (Clean & Corrupted) means that *lower gradient norm* leads to higher Test Accuracy (improved robustness). Once again, conflicting relationships are observed.

## 5.4 Different Robustness Definitions Requires Different Measures

We have now seen that conflicting relationships consistently arise between the measures and the different robustness definitions. This leads us to conclude that there is no one measure that is comprehensively reflective of robustness across all 4 definitions. Thus, we recommend that rather than finding a "one shoe fits all" measure, we should instead find a measure that is most representative for each respective definition of robustness. As we are more concerned with the classifier's performance on OOD data in the form of corruptions and adversarial examples, we focus on the Corruption Test Accuracy and ASR. Through our experiments, we found all decision-boundary based measures, *hessian eigenvalue*, and *weight gradient norm* to be most promising if we are concerned with the Corruption Test Accuracy. On the other hand, when concerned with the ASR, decision boundary-based measures prove to be most indicative. For decision boundary-based measures, we focus on *boundary thickness (PGD)*.

## 5.5 SHARPNESS OPTIMIZERS AND THEIR IMPACT ON SHARPNESS

To better understand the link between sharpness and robustness, we incorporated the use of sharpness optimizers into our training framework. Doing so allows us to obtain classifiers with different sharpness properties. This in turn enables us to discover evidence of correlations more easily between sharpness-based measures and robustness. To understand the impact that sharpness optimizers have, we plot the sharpness measures separately for the three cases (No sharpness optimizers, Sharpness-Aware-Minimization (SAM), and Adaptive-Sharpness-Aware-Minimization (ASAM)). From Figure 4, we see that for the same Model ID, classifiers trained with SAM consistently had the lowest sharpness. This was followed by classifiers trained with ASAM. Classifiers trained without sharpness optimizers had the highest sharpness value.

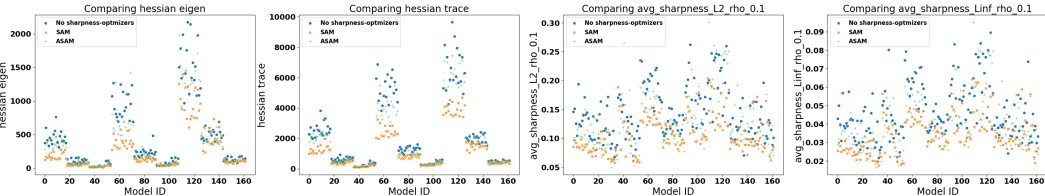

Figure 4: Scatter plots for sharpness-based measures for the different model IDs. The different model IDs correspond to different hyperparameter configurations. Within each model ID, the only difference in training configuration (hyperparameter setting) lies in the use of sharpness optimizers. Across all sharpness measures, classifiers trained with sharpness optimizers consistently yielded lower sharpness value. Between SAM and ASAM, SAM consistently obtained lower sharpness values.

Following the same approach, we plot the Generalization Gap, Test Accuracy (Clean & Corrupted), and ASR separately for the three cases involving different sharpness optimizers in Figure 5. We found that classifiers trained with sharpness optimizers consistently displayed higher robustness. Classifiers trained with SAM which have the lowest sharpness had the lowest Generalization Gap. They were also found to have the highest Test Accuracy (Clean & Corrupted), and the lowest ASR. On the other hand, classifiers with no sharpness optimizers had the highest Generalization Gap, lowest Test Accuracy (Clean & Corrupted), and highest ASR. This indicates their poor robustness.

Seeing as how using sharpness optimizers lead to lower sharpness and improved robustness, we might be tempted to correlate low sharpness with improved robustness. From our initial analysis, this is indeed a convincing argument as *hessian eigenvalue* has a correlation score $> |0.2|$ as seen in Figure 3.

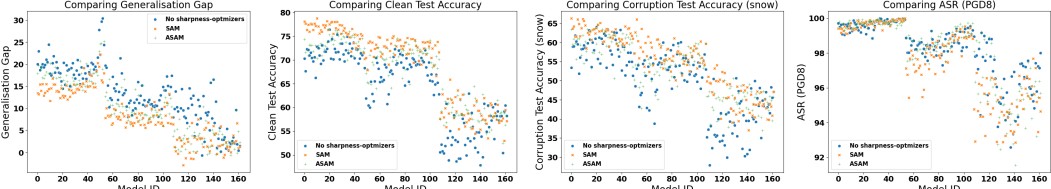

Figure 5: Scatter plots for the different robustness definitions for the different model IDs. As Corruption Test Accuracy and ASR involves aggregating data from the various corruptions and attack types, we reduced the scope of our analysis to make our analysis easier. We chose Corruption Test Accuracy (snow) to be representative of corruptions. For ASR, we chose the ASR of PGD ($\epsilon = 8/255$) to be representative. Across all definitions of robustness, classifiers trained with sharpness optimizers consistently yielded better robustness. Between SAM and ASAM, SAM consistently displayed better robustness.

However, further analysis finds that the improved robustness seemingly brought about by sharpness optimizers cannot be solely linked to sharpness. Other factors could have also contributed to the improved robustness. Batch size in particular plays a significant role in determining robustness. As

seen in Figure 6, within each sharpness optimizer case, clusters involving batch size are formed. These clusters contribute towards the negative correlation observed between Corruption Test Accuracy and *hessian eigenvalue*. Larger batch sizes tend to lead towards higher *hessian eigenvalue* (high sharpness), regardless of whether sharpness optimizers were utilized. Additionally, classifiers trained with smaller batch size are more likely to have low sharpness and high Corruption Test Accuracy. Given the significant role batch size plays, it would be incorrect to directly link low sharpness to higher Corruption Test Accuracy.

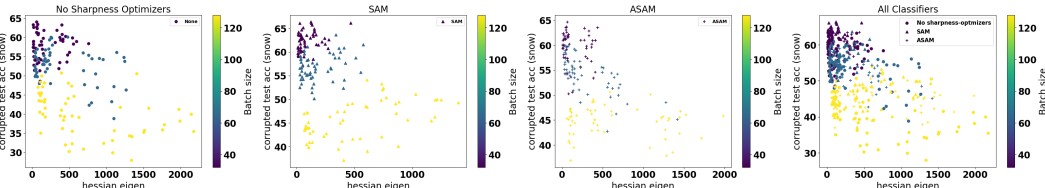

Figure 6: Scatter plots for *hessian eigenvalue* with Corruption Test Accuracy (Snow) based on the different sharpness optimizer settings. From the scatter plots, we observe that regardless of the sharpness optimizer setting, significant clusters involving batch size are formed

### 5.6 SHARPNESS OPTIMIZERS AND THEIR IMPACT ON BOUNDARY THICKNESS

We also analyze the impact that sharpness optimizers has on *boundary thickness*. We conclude that utilising sharpness optimizers tends to lead to thicker boundaries in most cases. However, as with the previous finding, the relationship between *boundary thickness* and robustness is heavily influenced by batch size. As seen from Figure 7, 3 distinct clusters corresponding to the different batch sizes $(32, 64, 128)$ are formed. These clusters contribute to the negative correlation obtained when correlating *boundary thickness* with Corruption Test Accuracy and ASR. Further analysis also found that within each cluster of batch size, learning rate also influences the relationship. The influence which learning rate holds is more apparent in clusters formed by larger batch sizes. As seen in Figure 7, especially in the clusters corresponding to batch size 128, classifiers with higher learning rate tend to have thicker boundaries.

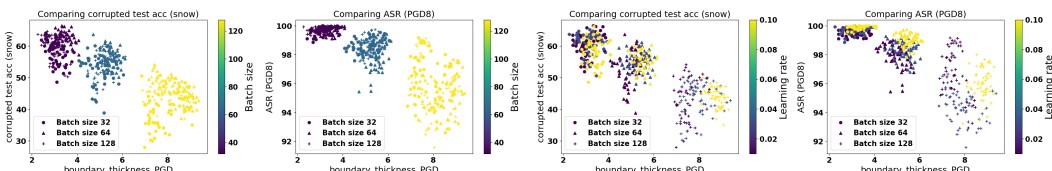

Figure 7: Scatter plots for *boundary thickness (PGD)* against Corruption Test Accuracy and ASR. Once again, distinctive clusters owing to the different batch sizes are formed. Additionally, learning rate forms further sub-clusters within each cluster.

## 6 CONCLUSION

In this work, we studied various measures and their ability to measure robustness. Through our experiments, we found that while certain measure appears as convincing candidates, inconsistencies were a common occurrence. While a measure might be reflective of a particular definition of robustness, it will imply a conflicting relationship with respect to another definition of robustness. This leads us to conclude that there is no one measure that is representative of robustness across all definitions. Thus, we suggest that rather than seeking a "one-shoe-fits-all" solution, we should instead use different measures to measure the different robustness definitions. As we are particularly interested in robustness from the perspective of corruptions and adversarial examples, we identified the *hessian eigenvalue* and *weight gradient norm* to be most representative of the Corruption Test Accuracy. For ASR, we identified *boundary thickness* to be most representative. In this work, we also studied the significance of sharpness in relation to robustness. Through our empirical studies,

we found that while there exists a relationship between sharpness and robustness, this relationship is tenuous. While low sharpness implies high Test Accuracy, it also implies high Generalization Gap. The relationship between sharpness and ASR is also weak. Furthermore, we found evidence of this relationship to be largely influenced by batch size. Analysis of other measures such as *boundary thickness* likewise yielded similar findings. We also found the effectiveness of *boundary thickness* to be influenced by the choice of hyperparameters such as batch size and learning rate. Nevertheless, through our analysis, we determined *boundary thickness* to be the most promising measure. It produced significant correlation scores across all the definitions of robustness. Despite the issues surfaced, we hope that *boundary thickness* can serve as a starting point in our bid to better understand robustness.

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

# A ALL RESULTS

## A.1 CLASSIFIERS WITHOUT ADVERSARIAL TRAINING

Table 1: Correlation analysis for the different measures against the different robustness definitions, for classifiers with no adversarial training.

| Measure name | Generalization Gap | Clean Test Accuracy | Corruption Test Accuracy | Attack Success Rate |
|---|---|---|---|---|
| num params | 0.008 | -0.048 | -0.003 | -0.006 |
| L2 | -0.101 | -0.053 | -0.029 | -0.048 |
| L2 init | -0.045 | -0.045 | -0.015 | -0.021 |
| path norm | 0.014 | -0.019 | -0.013 | -0.044 |
| frobenius over spectral | -0.142 | -0.143 | -0.088 | -0.106 |
| log spectral main term | 0.030 | 0.112 | 0.078 | 0.040 |
| log init spectral main term | 0.035 | 0.111 | 0.078 | 0.043 |
| log product of spectral | 0.011 | 0.075 | 0.056 | 0.034 |
| log product of spectral over margin | 0.034 | 0.111 | 0.078 | 0.043 |
| log product of frobenius | -0.028 | 0.007 | 0.017 | -0.017 |
| log product of frobenius over margin | -0.001 | 0.051 | 0.044 | -0.009 |
| log sum of spectral | 0.028 | 0.111 | 0.076 | 0.047 |
| log sum of spectral over margin | 0.060 | 0.156 | 0.103 | 0.059 |
| log sum of frobenius | -0.068 | 0.014 | 0.014 | -0.037 |
| log sum of frobenius over margin | -0.038 | 0.061 | 0.041 | -0.028 |
| sum of init spectral | 0.096 | 0.100 | 0.077 | 0.083 |
| sum of frobenius | -0.101 | -0.054 | -0.029 | -0.048 |
| sum of init frobenius | -0.045 | -0.045 | -0.015 | -0.020 |
| sparsity | 0.205 | 0.214 | 0.146 | 0.125 |
| inverse margin | 0.403 | 0.631 | 0.366 | 0.139 |
| boundary thickness PGD | -0.657 | -0.522 | -0.327 | -0.156 |
| boundary thickness FGSM | -0.653 | -0.524 | -0.327 | -0.164 |
| hessian eigenvalue | -0.169 | -0.349 | -0.209 | -0.104 |
| hessian trace | -0.104 | -0.245 | -0.146 | -0.085 |
| input grad norm | 0.265 | -0.220 | -0.081 | 0.105 |
| weight grad norm | -0.202 | -0.482 | -0.278 | -0.110 |
| avg sharpness L2 rho 0.05 | -0.085 | -0.263 | -0.147 | -0.035 |
| avg sharpness L2 rho 0.1 | -0.043 | -0.219 | -0.121 | -0.020 |
| avg sharpness L2 rho 0.2 | 0.059 | -0.089 | -0.047 | 0.006 |
| avg sharpness L2 rho 0.4 | 0.132 | 0.012 | 0.008 | 0.014 |
| avg sharpness Linf rho 0.1 | -0.076 | -0.264 | -0.147 | -0.032 |
| avg sharpness Linf rho 0.2 | -0.034 | -0.205 | -0.114 | -0.017 |
| avg sharpness Linf rho 0.4 | 0.085 | -0.054 | -0.026 | 0.012 |
| avg sharpness Linf rho 0.8 | 0.105 | -0.021 | -0.012 | 0.000 |

Table 2: Correlation analysis for the different measures against the different robustness definitions, for classifiers with adversarial training.

| Measure name | Generalization Gap | Clean Test Accuracy | Corruption Test Accuracy | Attack Success Rate |
|---|---|---|---|---|
| num params | 0.105 | -0.170 | -0.081 | 0.050 |
| L2 | 0.059 | -0.126 | -0.068 | 0.044 |
| L2 init | 0.101 | -0.113 | -0.057 | 0.042 |
| path norm | 0.143 | -0.094 | -0.054 | 0.063 |
| frobenius over spectral | -0.058 | -0.305 | -0.166 | 0.078 |
| log spectral main term | 0.027 | 0.016 | 0.013 | -0.014 |
| log init spectral main term | 0.037 | 0.014 | 0.013 | -0.012 |
| log product of spectral | 0.042 | 0.046 | 0.029 | -0.017 |
| log product of spectral over margin | 0.032 | 0.027 | 0.020 | -0.016 |
| log product of frobenius | 0.054 | -0.111 | -0.055 | 0.026 |
| log product of frobenius over margin | 0.036 | -0.139 | -0.070 | 0.030 |
| log sum of spectral | 0.020 | 0.098 | 0.057 | -0.033 |
| log sum of spectral over margin | 0.001 | 0.071 | 0.042 | -0.033 |
| log sum of frobenius | 0.035 | -0.094 | -0.050 | 0.024 |
| log sum of frobenius over margin | 0.012 | -0.127 | -0.067 | 0.029 |
| sum of init spectral | 0.125 | 0.221 | 0.128 | -0.049 |
| sum of frobenius | 0.059 | -0.126 | -0.068 | 0.044 |
| sum of init frobenius | 0.101 | -0.113 | -0.057 | 0.042 |
| sparsity | -0.009 | 0.152 | 0.094 | -0.067 |
| inverse margin | -0.201 | -0.219 | -0.123 | -0.004 |
| boundary thickness PGD | -0.599 | -0.521 | -0.278 | -0.007 |
| boundary thickness FGSM | -0.601 | -0.519 | -0.277 | -0.008 |
| hessian eigenvalue | -0.160 | -0.462 | -0.247 | 0.088 |
| hessian trace | -0.187 | -0.459 | -0.248 | 0.083 |
| input grad norm | -0.461 | -0.629 | -0.329 | 0.033 |
| weight grad norm | -0.331 | -0.509 | -0.275 | 0.058 |
| avg sharpness L2 rho 0.05 | -0.266 | -0.608 | -0.318 | 0.082 |
| avg sharpness L2 rho 0.1 | -0.243 | -0.591 | -0.309 | 0.085 |
| avg sharpness L2 rho 0.2 | -0.118 | -0.505 | -0.266 | 0.101 |
| avg sharpness L2 rho 0.4 | 0.175 | -0.242 | -0.130 | 0.116 |
| avg sharpness Linf rho 0.1 | -0.261 | -0.604 | -0.316 | 0.082 |
| avg sharpness Linf rho 0.2 | -0.233 | -0.585 | -0.307 | 0.087 |
| avg sharpness Linf rho 0.4 | -0.063 | -0.465 | -0.246 | 0.107 |
| avg sharpness Linf rho 0.8 | 0.206 | -0.214 | -0.116 | 0.117 |

# B  ADDITIONAL EXPERIMENTS

## B.1  CONSIDERING CLASSIFIERS WITH ADVERSARIAL TRAINING

The vulnerability of DNNs to adversarial examples has been well demonstrated. This indicates the need for appropriate defences to deter attackers. A common defensive technique to increase the robustness of DNNs against adversarial attacks is to perform adversarial training (Goodfellow et al., 2014). By incorporating adversarial examples into the training dataset, the DNN would be able to learn the features corresponding to the adversarial examples and still yield correct outputs. Given the popularity of adversarial training, it leads us to question if the previous relationships

learnt are also applicable to classifiers which have undergone adversarial training. To investigate this, we follow the same approach as before. However, we now only consider classifiers which have undergone adversarial training. To do so, we additionally trained 486 classifiers with adversarial training. Thereafter, we repeated our experiments as before, performing the measurements and correlating them against the 4 robustness definitions. Our experiments show that the previously identified relationships do not always hold when we factor in adversarial training. For the two scenarios of without and with adversarial training, for the same measure, different behaviors can be observed. Different behaviors include scenarios where the relationship learnt is flipped. We also found some cases where the measures lose their ability to reflect robustness. Like in the previous study, we split our analysis of the measures into 4 categories.

**Complexity-based Measures.** In the study where classifiers were trained without adversarial training, among all complexity-based measures, only *sparsity* obtained a correlation score $> |0.2|$ for some robustness definitions. However, as seen in Figure 8., for classifiers trained with adversarial training, the effectiveness of *sparsity* was not as pronounced. Instead, measures like *Frobenius_over_spectral* and *sum_of_init_spectral* appeared more convincing. Additionally, while high *sparsity* previously implied high ASR, high *sparsity* now implies low ASR. These differences in relationships indicate that the relationship learnt in the previous study is not directly applicable to classifiers which have undergone adversarial training. Different behaviors are observed.

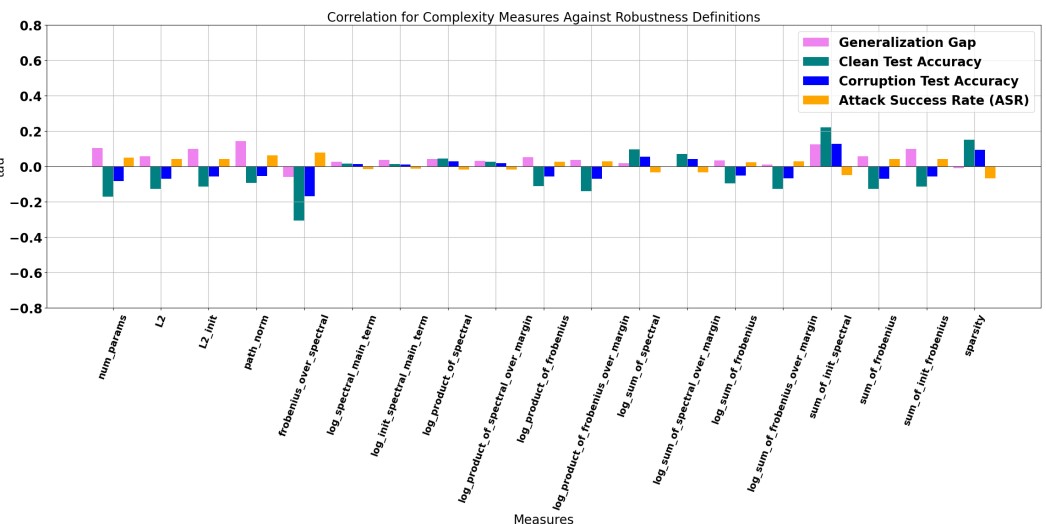

Figure 8: Correlation scores when correlating complexity-based measures against the different definitions of robustness for classifiers which have undergone adversarial training.

**Decision boundary-based Measures.**

- **Inverse margin.** As seen in Figure 9., the correlation between *inverse margin* and the robustness definitions of Generalization Gap and Test Accuracy (Clean & Corrupted) are all negative. Furthermore, the correlation with ASR is close to 0, indicating that there is no relationship between *inverse margin* and the ASR. This is opposed to the case in Figure 2., which displays positive correlation scores between *inverse margin* and all 4 definitions of robustness. The change in polarities for the correlation indicate that the relationship learnt has been entirely flipped. While classifiers without adversarial training show that thick margins imply low Generalization Gap, classifiers with adversarial training show that thick margins imply high Generalization Gap.

- **Boundary thickness.** While the correlation relationship of *boundary thickness* with Generalization Gap and Test Accuracy (Clean & Corrupted) is similar to that obtained when no adversarial training is considered, we found that the correlation score with ASR dropped significantly. As seen in Figure 9., the correlation score between *boundary thickness* and ASR is essentially 0. This contrasts with the case in Figure 2, where the correlation score between *boundary thickness* and ASR is just below $|0.2|$. This indicates that for classifiers

with adversarial training, *boundary thickness* is unable to act as a measure of robustness in terms of ASR.

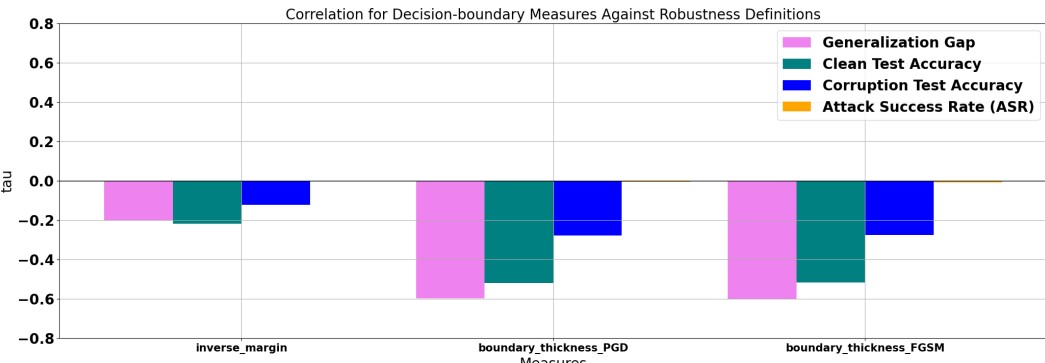

Figure 9: Correlation scores when correlating decision boundary-based measures against the different definitions of robustness for classifiers which have undergone adversarial training.

**Sharpness-based measures.** Compared against the previous study, we found that classifiers with adversarial training yield stronger relationships when correlating sharpness-based measures against the Generalization Gap and Test Accuracy (Test & Corrupted). We also found that their relationships with ASR are flipped. As seen in Figure 10., we now obtain a positive relationship with ASR instead of a negative one. This indicates that lower sharpness means lower ASR. This makes sharpness a good candidate for consideration when considering classifiers with adversarial training, as it displays a consistent relationship with robustness when considering robustness in terms of Test Accuracy (Clean & Corrupted) and ASR.

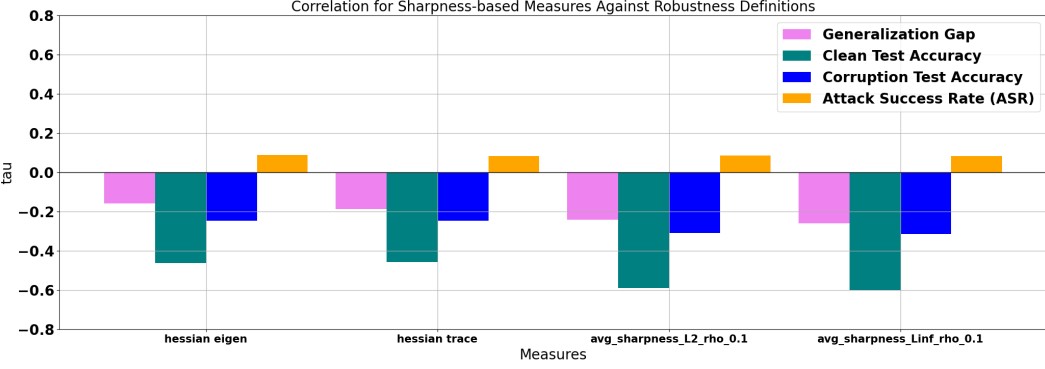

Figure 10: Correlation scores when correlating sharpness-based measures against the different definitions of robustness for classifiers which have undergone adversarial training.

**Gradient-based measures.**

- **Input-gradient norm.** Compared to the study where adversarial training is not considered, the correlation of *input-gradient norm* with Generalization Gap now flips from positive to negative. This implies that high *input gradient norm* now leads to low Generalization Gap. While there was no change in polarity for the relationship with Test Accuracy (Clean & Corrupted), we found that this relationship was weaker for classifiers with adversarial training. Relationship with ASR was observed to be similar too, albeit slightly weaker.

- **Weight-gradient norm.** When comparing the relationship obtained between classifiers without and with adversarial training, we found the polarity of the correlation scores to remain the same for Generalization Gap and Test Accuracy (Clean & Corrupted). However, the strength of correlation for the case with adversarial training was found to be

stronger. Besides this, we also observed the switch in polarity of correlation scores for ASR compared to that of the previous study. While the previous relationship between *weight-gradient norm* and ASR was negative, the relationship obtained is now positive.

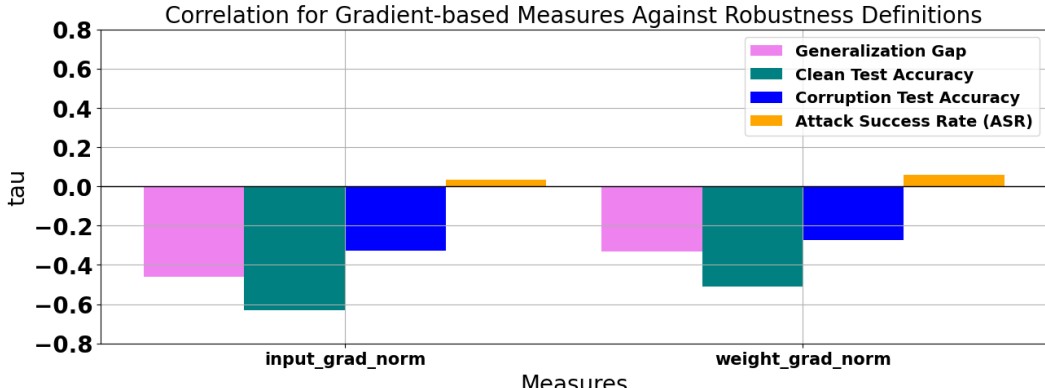

Figure 11: Correlation scores when correlating gradient-based measures against the different definitions of robustness for classifiers which have undergone adversarial training.

### B.1.1 RECOMMENDED MEASURES FOR CLASSIFIERS WITH ADVERSARIAL TRAINING

We previously recommended several measures for both Corruption Test Accuracy and ASR. However, these relationships were learnt only from classifiers without adversarial training. We now recommend measures based on classifiers which have undergone adversarial training. For these classifiers, we found that *boundary thickness*, all sharpness-based measures, and all gradient-based measures were indicative of robustness in terms of Corruption Test Accuracy. On the other hand, when concerned with the ASR, only sharpness-based measures appeared most indicative of robustness. This means that unlike the case where no adversarial training is considered, we did find evidence of a measure that is reflective of robustness in terms of both Corruption Test Accuracy and ASR. In particular, sharpness-based measures such as the *hessian eigenvalue* displayed this.

### B.1.2 COMPARING RELATIONSHIPS FOR CLASSIFIERS WITHOUT AND WITH ADVERSARIAL TRAINING

Upon comparing the relationships learnt from classifiers without and with adversarial training, we noticed a few considerable differences. We summarize the key differences into 3 points.

**Some measures which are indicative of robustness for classifiers without adversarial training are not indicative of robustness for classifiers with adversarial training.** An example of this is *boundary thickness*. While *boundary thickness* was found to a promising measure of robustness in terms of ASR in the scenario where adversarial training is not considered, this does not hold true for classifiers with adversarial training. Rather than *boundary thickness* which yielded a correlation score close to 0, sharpness-based measures were found to best reflect robustness in terms of ASR.

**Some measures which are indicative of robustness for classifiers with adversarial training are not indicative of robustness for classifiers without adversarial training.** From our experiments, we observed that there exist more measures that are indicative of robustness (Corruption Test Accuracy) when considering classifiers with adversarial training. This includes the *input-gradient norm* and the other sharpness-based measures besides the *hessian eigenvalue*. These same measures were not able to capture the robustness relationships for classifiers without adversarial training.

**Indicative measures shared by both scenarios carry different meanings.** While both scenarios without and with adversarial training shared similar measures which could be indicative of robustness (correlation score $> |0.2|$), certain relationships imply opposing meanings due to the

phenomenon of flipped relationships. This is especially so in the case of *inverse margin* when concerned with Corruption Test Accuracy, and all sharpness-based measures when concerned with the ASR. While *inverse margin* had a correlation score $> |0.2|$ with Corruption Test Accuracy in both scenarios, this was a positive relationship in the case for classifiers without adversarial training and a negative relationship when considering classifiers with adversarial training. A positive relationship indicates that low *inverse margin* (large margins) leads to lower Corruption Test Accuracy while a negative relationship indicates that low *inverse margin* (large margins) leads to higher Corruption Test Accuracy.

### B.1.3    Recommended Measures for Classifiers without and with Adversarial Training

Considering the fact that the relationships in one scenario may not hold in the other, we now recommend common measures that are applicable in both scenarios. Among the measures studied for Corruption Test Accuracy, we found that *boundary thickness*, *hessian eigenvalue*, and *weight-gradient norm* are promising measures when considering both scenarios. On the other hand, for ASR, none of our studied measures can reflect robustness when considering both scenarios.

## C    Additional Analysis

### C.1    Significance of the Different Measures

Putting aside the issue of inconsistencies that we surfaced earlier, several viable measures for each robustness definitions do exist. We are interested to understand which measure contributes most to robustness. However, given that the link between these measures and robustness is still not well understood, and the existence of inconsistencies, this is a difficult task. Thus, we decided to take an unconventional approach to instead offload this task to an auxiliary model.

In particular, we trained a set of 4 different regression models to predict each robustness definition (Generalization Gap, Clean Test Accuracy, Corruption Test Accuracy, and ASR). These models take in the viable measures as input and the different robustness definitions as output. In most cases, we deem a measure to be viable if its correlation score is $> |0.2|$. After training these models, we utilize SHapley Additive exPlanations (SHAP) values to explain how important each feature (measures) is to the model when it predicts robustness. SHAP values indicates to us each features contribution to the predicted output. To aid our analysis of SHAP values, we utilize Beeswarm plots. Beeswarm plots tell us the relative importance of the features and their actual relationships with the predicted outcomes.

Through this analytical process, we offload the problem of understanding how significant a measure is with respect to each other to the regression model. While this method has its flaws, we believe that it gives us an indication of which measure holds greater significance.

### C.1.1    Significance of the Measures in predicting the generalization Gap

To understand how important each measure is when predicting the Generalization Gap, we train a regression model to predict the Generalization Gap. We used measures which had correlation scores $> |0.2|$ to train this model. Thereafter, we calculated the SHAP values of each feature. From Figure 12, we see that among the features (measures), *boundary thickness* is the most significant feature when predicting the Generalization Gap. This is followed by gradient-based measures. For *boundary thickness*, we observed dense clusters of low *boundary thickness* (in blue) with positive SHAP values. On the other hand, data points with high *boundary thickness* (in red) are more spread out and have negative SHAP values. This indicates that the negative correlation between *boundary thickness* and Generalization Gap is strong. As *boundary thickness* increases, Generalization Gap decreases. This supports our earlier finding.

### C.1.2    Significance of the Measures in predicting the Clean Test Accuracy

Likewise for Clean Test Accuracy, we trained a regression model to predict it using measures with correlation scores $> |0.2|$ as input. Performing the same analysis also results in *boundary thickness*

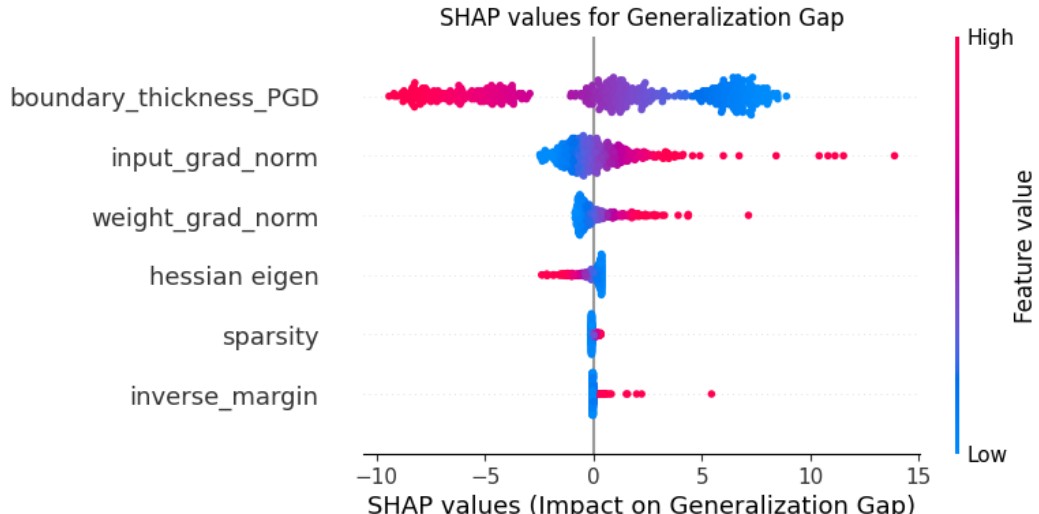

Figure 12: SHAP Beeswarm plot for predicting the Generalization Gap.

and gradient-based measures being deemed as the most important measures for predicting Clean Test Accuracy. The significance of *boundary thickness* can be seen from Figure 13, where *boundary thickness* is ranked at the top, indicating that it is the most significant measure contributing to the model's prediction. In this case, our SHAP analysis indicates that high *boundary thickness* contributes to lower Clean Test Accuracy. This again is a similar observation to what we had in the earlier sections.

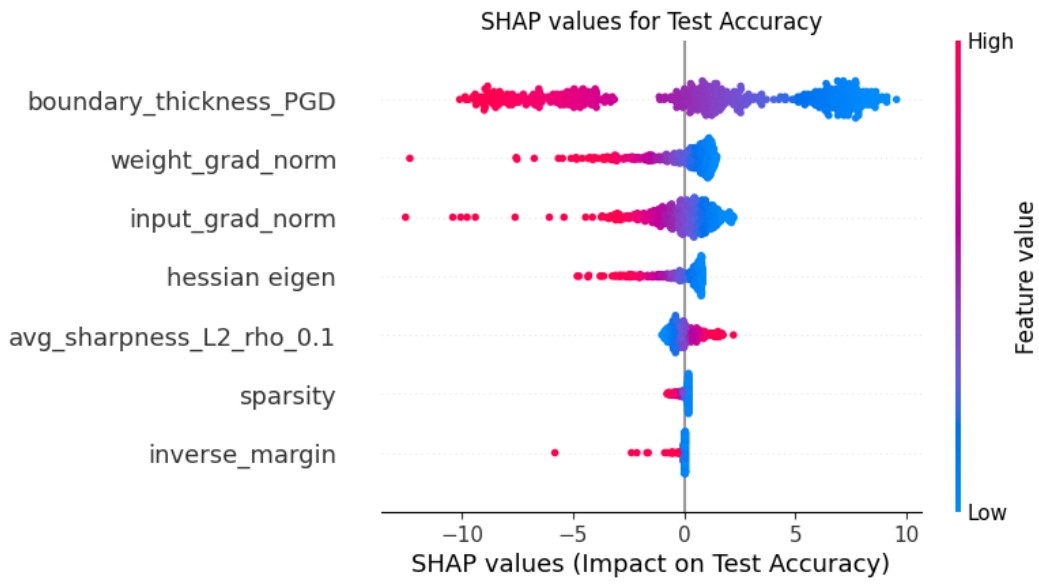

Figure 13: SHAP Beeswarm plot for predicting the Clean Test Accuracy.

### C.1.3 SIGNIFICANCE OF THE MEASURES IN PREDICTING THE CORRUPTION TEST ACCURACY

Following the same procedure for Corruption Test Accuracy yields the same results and findings. As seen in Figure 14, when trained to predict the Corruption Test Accuracy, among the measures of interest, *boundary thickness* is found to be the most significant measure to the model.

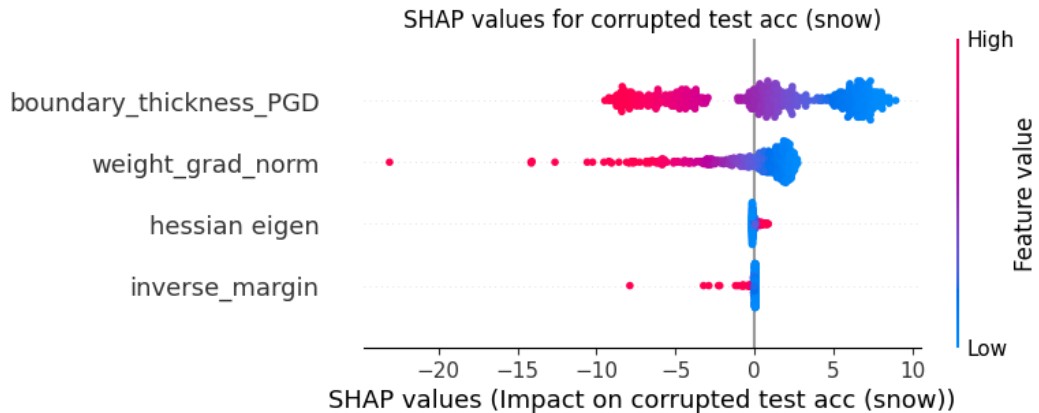

Figure 14: SHAP Beeswarm plot for predicting the Corruption Test Accuracy.

### C.1.4 SIGNIFICANCE OF THE MEASURES IN PREDICTING THE ATTACK SUCCESS RATE

When analyzing the significance of measures in predicting the ASR, we first note that none of the measures have correlation scores $> |0.2|$. Thus, to train our regression models to predict ASR, we relaxed our previous condition and instead used measures with correlation scores $> |0.1|$ as input. Performing the same analysis results in *boundary thickness* being the most important feature. We found that as *boundary thickness* increases, ASR decreases. This relationship agrees with our previous finding.

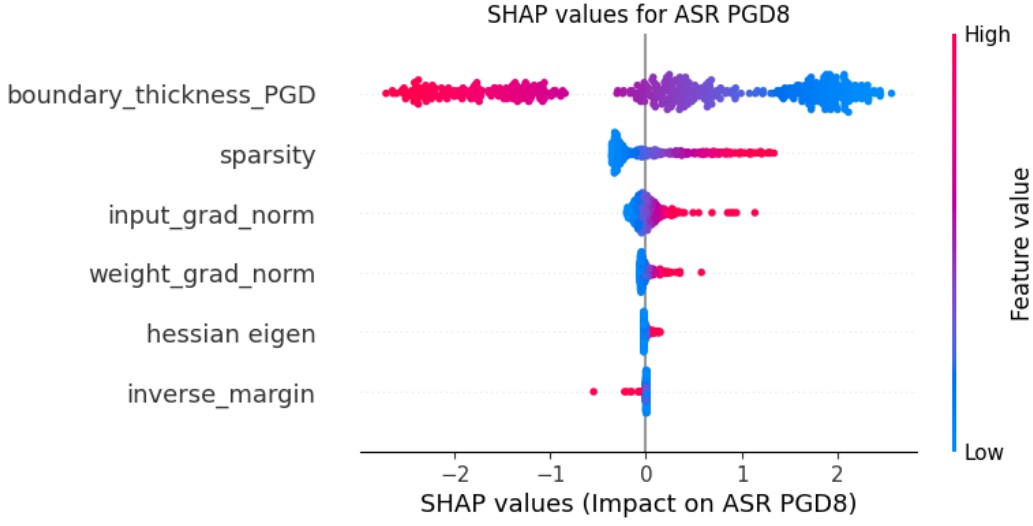

Figure 15: SHAP Beeswarm plot for predicting the ASR.

### C.1.5 IMPORTANCE OF BOUNDARY THICKNESS

Our SHAP analysis came to the conclusion that *boundary thickness* is the most indicative measure of robustness. However, it is important to note that these findings might be marred by the performance of the regression models themselves. While we did verify the trained models to have low mean square error and high R2 values, the ability of these models to predict values which are truly representative of the different definitions of robustness is still questionable. Additionally, inconsistencies still arises, while thick boundary thickness means low Generalization Gap and low ASR, it also leads to low Test Accuracy (Clean & Corrupted). Despite these issues, we believe that this analysis serves to identify the significant role which *boundary thickness* plays in predicting robustness.

## D DEFINITIONS OF MEASURES

We provide a more in depth explanation on the measures, the intuition behind them, and what they represent. We also provide their mathematical formulations where relevant.

### D.1 COMPLEXITY MEASURES

These measures attempt to capture how complex the learnt function (DNN weight matrix) is. In general, the less complex the learnt function is, the more robust the DNN is. A majority of our complexity-based measures based on the Spectral and Frobenius norm are adopted from Dziugaite et al. (2020), which modified measures introduced by Jiang et al. (2019). For these measures, we also calculated variants with respect to the initialisation value of the initial weight matrix at time-step 0. We represent these variants with the term *init*. We let $W_i$ represent the weight tensor belonging to layer $i$ of the DNN, $d$ represent the depth of the DNN, and $m$ represent the train dataset size.

**Number of parameters.** Calculates the number of learnable parameters the DNN has.

$$num\_params = \sum_i^d k_i^2 c_{i-1}(c_i + 1) \tag{1}$$

At each layer $+i$, we have a $k_i$ x $k_i$ kernel and $c_i$ filters. Given the same network architecture, this measure only differs when varying the width and depth of the DNN.

**Path-norm.** Takes the summation of the product of the weights along all paths of the DNN, from an input neuron to an output neuron. This can be calculated by taking the sum of outputs of a DNN with squared weights $f(W^2)$ when passing in a vectors of ones as inputs.

$$path\_norm = \sum_i f_{w^2}(1) \tag{2}$$

**Spectral norm.** Calculated using the methods introduced by Sedghi et al. (2018), the spectral norm gets the maximum singular vector of the weight matrix. We denote the spectral norm as $\|W_i\|$.

- Log spectral main term

$$log\_spec\_main\_term = log\sqrt{\frac{\prod_{i=1}^d \|W_i\|_2^2 \sum_{j=1}^d \frac{\|W_j\|_F^2}{\|W_j\|_2^2}}{\gamma^2 m}} \tag{3}$$

- Log init spectral main term

$$log\_init\_spec\_main\_term = log\sqrt{\frac{\prod_{i=1}^d \|W_i\|_2^2 \sum_{j=1}^d \frac{\|W_j - W_j^0\|_F^2}{\|W_j\|_2^2}}{\gamma^2 m}} \tag{4}$$

- Log product of spectral

$$log\_prod\_of\_spec = log\sqrt{\frac{\prod_{i=1}^d \|W_i\|_2^2}{m}} \tag{5}$$

- Log product of spectral over margin

$$log\_prod\_of\_spec\_over\_margin = log\sqrt{\frac{\prod_{i=1}^{d} \|W_i\|_2^2}{\gamma^2 m}} \qquad (6)$$

- frobenius over spectral

$$frob\_over\_spec = log\sqrt{\frac{\sum_{i=1}^{d} \frac{\|W_i\|_F^2}{\|W_i\|_2^2}}{m}} \qquad (7)$$

- Log sum of spectral

$$log\_sum\_of\_spec = log\sqrt{\frac{d(\prod_{i=1}^{d} \|W_i\|_2^2)^{\frac{1}{d}}}{m}} \qquad (8)$$

- log sum of spectral over margin

$$log\_sum\_of\_spec\_over\_margin = log\sqrt{\frac{d(\frac{\prod_{i=1}^{d} \|W_i\|_2^2}{\gamma^2})^{\frac{1}{d}}}{m}} \qquad (9)$$

- sum of init spectral

$$sum\_of\_init\_spec = \sqrt{\frac{\sum_{i=1}^{d} \|W_i - W_i^0\|_2^2}{m}} \qquad (10)$$

**Frobenius norm.** The square root of the sum of the absolute squares of the elements in the weight matrix. We represent the Frobenius norm as $\|W_i\|_F^2$.

- Log product of frobenius

$$log\_spec\_main\_term = log\sqrt{\frac{\prod_{i=1}^{d} \|W_i\|_F^2}{m}} \qquad (11)$$

- Log product of frobenius over margin

$$log\_prod\_of\_spec\_over\_margin = log\sqrt{\frac{\prod_{i=1}^{d} \|W_i\|_F^2}{\gamma^2 m}} \qquad (12)$$

- Log sum of frobenius

$$log\_sum\_of\_frob = log\sqrt{\frac{d(\prod_{i=1}^{d} \|W_i\|_F^2)^{\frac{1}{d}}}{m}} \qquad (13)$$

- Log sum of frobenius over margin

$$log\_sum\_of\_frob\_over\_margin = log\sqrt{\frac{d(\frac{\prod_{i=1}^{d} \|W_i\|_F^2}{\gamma^2})^{\frac{1}{d}}}{m}} \qquad (14)$$

- sum of frobenius

$$sum\_of\_frob = \sqrt{\frac{\sum_{i=1}^{d} \|W_i\|_F^2}{m}} \qquad (15)$$

- sum of init frobenius

$$sum\_of\_init\_frob = \sqrt{\frac{\sum_{i=1}^{d} \|W_i - W_i^0\|_F^2}{m}} \qquad (16)$$

**Sparsity.** The ratio of elements in the weight matrix which has values below a threshold value. A higher sparsity means that more elements in the weight matrix falls below the threshold value. This indicates a less complex DNN.

$$sparsity = \frac{\sum_{i=1}^{d} 1[W_i < threshold]}{|W|} * 100\% \qquad (17)$$

## D.2 Decision Boundary Measures

These measures provide some insight into what is the distance or the perturbation that is required to cross over the class boundaries. Intuitively, the harder it is to cross over the decision boundaries, the more robust the DNN is.

**Inverse margin.** The margin $\gamma$ between class boundaries for each data point is defined as the difference between the top-2 logit values. To get a margin value that is representative of the entire train dataset, we first calculate the margins for all examples in the train dataset. Thereafter, we take the 10th percentile of the calculated margins as the final margin value. Following this, the inverse margin is calculated by taking the reciprocal of the square of the final margin value.

$$inverse\_margin = \frac{1}{\gamma^2} \tag{18}$$

**Boundary thickness.** Introduced by Yang et al. (2020), *boundary thickness* measures the average distance (L2 norm) between the set of adversarial examples and the set of natural examples. This can be seen as a generalized form of margin which only takes the worst case (difference between the top-2 logit values). However, when calculating *boundary thickness*, we randomly sample $n$ times from a mixup of the adversarial and natural examples and calculate the distance between the selected points. This aims to capture how thick the boundary between the set of adversarial and natural examples is.

## D.3 Sharpness Measures

These aim to estimate the sharpness of the loss landscape. This follows from works that attempt to link the robustness of DNNs to the sharpness of loss landscapes.

**Hessian measures.** These measures are based on the Hessian matrix of a DNNs loss function with respect to its parameters. This contains second-order partial derivatives which provides information on the curvature of the loss landscape. In our experiments, we measure both the maximum eigenvalue of the Hessian and the trace of the Hessian. These 2 measures capture different aspects of the loss landscape.

- **Hessian eigenvalue.** The maximum eigenvalue of the Hessian matrix indicates the direction of the largest curvature (worst-case). The larger in magnitude these values are, the sharper the loss landscape is. Additionally, while a positive value indicates the loss landscape at the point is concave upwards, a negative value indicates that the loss landscape is concave downwards.
- **Hessian trace.** The *hessian trace* is the sum of all Hessian eigenvalues. This measures the overall curvature of the loss landscape. The larger these values are, the sharper the loss landscape is.

**Average sharpness.** Estimates sharpness by taking the difference in loss values between a DNN with injected noise and without noise. Noisy DNNs are constructed by injecting noise into the original DNNs parameters at random. Large differences in loss values indicates a sharp loss landscape. We conduct a few variants of this measure. In particular, we explored adding noise with the L2 and L-infinity constraints. We also varied the variance in which noise was added to the DNNs when constructing noisy DNNs.

## D.4 Gradient Measures

**Input-gradient norm.** The vulnerability of DNNs has been linked to the noisiness of the input gradients. Works have found that through regularizing *input-gradient norms* (Ross & Doshi-Velez, 2018), it leads to a smoothing effect which increases the robustness of DNNs to adversarial examples. Thus, by measuring the *input-gradient norm* of DNNs, we hope to link it to the robustness of DNNs.

$$input\_gradient\_margin = \mathbb{E}_{x,y}[\|\nabla_x \mathcal{L}_{CE}(f(x,W),y)\|_2] \tag{19}$$

**Weight-gradient norm.** Regularizing the *weight-gradient norm* has a similar effect to obtaining a low local Lipschitz (Zhao et al., 2022), where a low local Lipschitz is linked to obtaining a flat minimum. Seeing as how flat minima has been linked to increased robustness, we hope to find some association between *weight-gradient norm* and robustness too.

$$weight\_gradient\_margin = \mathbb{E}_{x,y}[\|\nabla_W \mathcal{L}_{CE}(f(x,W),y)\|_2] \tag{20}$$

# E  IMPLEMENTATION DETAILS

## E.1  IMAGE CLASSIFIER TRAINING CONFIGURATIONS

To discover convincing measures of robustness, we require a pool of well-trained classifiers with vastly different robustness behaviors. In our experiments, we trained multiple ResNet image classifiers under different hyperparameter configurations with no augmentations considered. We trained till convergence (cross-entropy 0.01) and repeated each experiment 3 times, each with different weight initialization values. Performing this resulted in 486 different hyperparameter configurations and a total of 1458 classifiers. We detail the different hyperparameter configurations below.

1. Depth: Varies between depths of {*ResNet-18, ResNet-34*}.
2. Dropout: Varies between dropouts of {*0, 0.25, 0.50*}.
3. Batch size: Varies between batch sizes of {*32, 64, 128*}.
4. Optimizers: Varies between these optimizers {*"SGD", "SGD-SAM", "SGD-ASAM"*}.
5. Learning Rate: Varies between learning rates of {*0.01, 0.032, 0.1*}.
6. Weight Decay: Varies between weight decays of {*0, 0.0001, 0.0005*}.

For the study done on classifiers with adversarial training, we reuse the same hyperparameter configurations as above with the exception of varying the optimizer. This results in 162 different hyperparameter configurations and a total of 486 classifiers. We performed adversarial training by employing the learning objective introduced by Goodfellow et al. (2014).

## E.2  MEASURE CONFIGURATIONS

When performing the measures, there exist certain hyperparameters to be set too. In this section, we detail the settings we used when conducting the measures.

To calculate *sparsity* of the weight matrix, we took the threshold value to be 1% of the maximum element of the weight matrix. The more elements in the weight matrix that falls below the threshold value, the higher the *sparsity*.

*Boundary thickness* calculates the L2 distance between the set of natural and adversarial images. We calculate *boundary thickness* in two ways, with respect to both PGD and FGSM attacks. We term them as *boundary_thickness_PGD* and *boundary_thickness_FGSM*.

When measuring *average sharpness*, we introduce some variations during the measurements. Particularly, when creating the noisy classifiers, we add noise with the L2 and L-infinity constraints. We also vary the amount of noise that is injected into the original weights by varying the variance of added noise. We term the hyperparameter that controls the amount of injected noise as rho. In our experiments, while we explored multiple rho values, we eventually only considered the scenario where rho is set to 0.1 for our results and analysis.

1. L2 average sharpness: {*0.05, 0.1, 0.2, 0.4*}
2. L-infinity average sharpness: {*0.1, 0.2, 0.4, 0.8*}

For corruptions and adversarial examples, there exists multiple variants and thus numerous readings within each of them. This includes 14 Corruption Test Accuracy values per trained classifier corresponding to the 14 corruptions introduced. For adversarial examples, as we have 2 different attacks

algorithms and 3 attack budget settings per attack algorithm, this cumulates to 6 ASR readings for each trained classifier. Given the large number of readings we have for these two robustness definitions, we decided to aggregate the Corruption Test Accuracies for all 14 corruptions together when performing correlation analysis for Corruption Test Accuracy. Likewise, for ASR, we aggregate the ASR for all attack variants together when performing correlation analysis for ASR.

