# OpenReview forum: "Measuring and Improving Robustness of Deep Neural Networks"
_ICLR.cc/2025/Conference — ICLR 2025 Conference Withdrawn Submission_

### Official Review · Reviewer_m2Q7 · 2024-10-29

**Soundness:** 2
**Presentation:** 3
**Contribution:** 2
**Rating:** 3
**Confidence:** 4

**Summary:**

This paper studies multiple measures and their capability to measure robustness. Inconsistency is broadly observed when testing the correlation of the measure to tested robustness. The suggestion of using different measures for different robustness definitions is proposed. Several representative measures are identified for individual robustness definitions, and the impact of sharpness-aware optimization is also considered.

**Strengths:**

The tackled problem is important, as the deployment of DNNs in practice requires robustness to some extent. Currently, a proper measure for estimating robustness is lacking and would be crucial for safe and generalizable deployment. The paper conducts experiments across a broad range of candidate measures and found none of them can serve as a ready-to-be-used choice.

**Weaknesses:**

My concern on weaknesses are as follows.
- Only a narrow range of models is considered. Throughout the paper, resnet-18 and resnet-34 plus a *single* dataset are used. Though the study covers a nice suite of measures, it lacks an analysis on other model structures (e.g., ViT) and other datasets.
- Definition of robustness is a bit unclear. The robustness to common corruptions and to adversarial examples are widely adopted in previous studies. However, whether it is proper to call the clean test accuracy and the generalization gap as "robustness" remains elusive. Following this ambiguity, the claim on inconsistency of measures against these definitions does not necessarily infer that these measures are not informative for OOD/adversarial robustness.
    - Besides, there are multiple datasets that are suitable, and serve as benchmark datasets to measure OOD accuracy, e.g., CMNIST/PACS/Waterbird, but they are not considered in this paper.
- Some key claims on inconsistency is not necessarily "inconsistent". For example, a measure yielding high test accuracy and high generalization gap simultaneously does not necessarily imply an inconsistency, since a higher train accuracy can explain why these two phenomenons could happen.
- Only sharpness-aware minimization is considered out of ERM. There are other candidate algorithms for better OOD generalization, e.g., invariant rist minimization, but they are not considered in this paper.

**Questions:**

Following weaknesses, I have the following questions.
- For definition of robustness. I would be happy to see the OOD test accuracy on benchmarking datasets are taken into consideration on top of the common corruption studied in this paper, since it would make the discussion on robustness more complete. Besides, a short discussion on why clean accuracy is suitable for testing is necessary since it is not commonly discussed in the context of robustness.
- For algorithms considered. I think taking other OOD generalization algorithms into consideration would benefit the completeness of the paper as well, since now only SAM is launched in this study.
- For model and dataset. Covering a broader range of models would make the claim in the paper more trustworthy.
- Why a correlation score with > |0.2| is considered as informative throughout the paper?

---

### Official Review · Reviewer_oTfK · 2024-10-29

**Soundness:** 3
**Presentation:** 2
**Contribution:** 1
**Rating:** 3
**Confidence:** 4

**Summary:**

The paper compares various robustness metrics which have been used in the literature on adversarially robust networks and learning with corruption type setups. Their goal is to find a metric (or set of metrics) that accurately quantify the robustness of a network, whether that be complexity-based, sharpness-based, or a traditional robustness metric (adversarial accuracy, etc). In targeting this, they compare various networks across each of these criteria and assess the concordance (or lack thereof) of the different notions of robustness. They later show that training with e.g. sharpness-aware optimizers that target low sharpness (which was shown to correlate with robustness) leads to more robust classifiers.

**Strengths:**

-- The paper is clear and well-written. The figures are clear and typeset well.

-- I find the overall goals of unifying various robustness metrics and assessing their concordance to be a sound one. While for example it is known that l-infinity defenses will protect against e.g. l-2 attacks to some extent, and vice versa, existing robustness papers typically choose a single or a few narrow notions of robustness to protect against. The paper therefore serves a valid empirical purpose.

-- The paper is relatively comprehensive. They compare complexity measures, sharpness measures, measures on the margin of decision boundaries, etc. This appears to be a heavy practical lift for which the findings can be of service to the community.

**Weaknesses:**

-- My main concern is that I find the overall novelty of the paper to be low. It is comprehensive from an empirical point of view, but the overall novelty appears limited as many of these relationships between different robustness metrics have been studied individually in prior works.

-- The paper shows that training with sharpness-aware optimizers leads to more robust classifiers, it doesn't provide theoretical analysis explaining why this relationship exists. Ditto for complexity measures -- these are supposed to correlate with generalization performance -- can there be an interesting link to robustness and generalization from a theoretical standpoint? I find the paper would add clutter to the literature rather than clarity.

-- Some of the metrics seem rather redundant and only serve to add noise to the paper. For example, on pg. 21 they include log sum of frobenius norms, log sum of frobenius norms over margin and sum of frobenius norms. They feel redundant, and if they are not, the authors should explain why all of these metrics are included.

**Questions:**

-- Why train with so many architectures and hyperparameter configurations? I would have liked to see an experiment where they control the architecture and give more analysis in a single case, as I find the aggregared results to be a little hard to follow.

-- I would suggest that the authors trim the number of metrics they include in future versions. As a reader I find it hard to follow and somewhat disorienting when there are multiple categories of metrics, each with similar meaning, for which I need to interpret the correlations.

---

### Official Review · Reviewer_t5Q7 · 2024-10-29

**Soundness:** 3
**Presentation:** 2
**Contribution:** 1
**Rating:** 3
**Confidence:** 3

**Summary:**

The authors study the correlation between different existing measures of robustness with 4 robustness definitions: generalization gap, test accuracy on clean data, test accuracy on corrupted data, and attack success rate.  They find that existing metrics are generally not correlated with all robustness definitions or exhibit contradictory relationships (positive correlation with test accuracy but also positive correlation with generalization gap).  They then investigate the robustness of training methods which regularize sharpness and find that the correlations between sharpness and robustness are influenced largely by the choice of training hyperparameters.

**Strengths:**

- large scope of experiments: the authors provide an in-depth analysis and discussion of correlations between many different metrics and the 4 robustness definitions
- writing is clear

**Weaknesses:**

- Presentation of figures: the plots of correlation can be a bit difficult to interpret because some metrics are designed so that smaller values of that metric should be more robust while others are designed so that larger values of that metric are more robust.  It would be a lot easier to interpret and compare correlation plots if metrics are all plotted so that positive correlation means better robustness.  This would mean that for metrics that are designed so that smaller values means better robustness, the authors plot the correlation between something like -1 * metric with the robustness measures instead.  Similarly, I think it would help with presentation if the authors applied this to generalization gap as well since smaller generalization gap means better robustness.
- Motivation: The authors found that hessian eigenvalue and weight gradient norm to be most representative of corruption accuracy and boundary thickness to be most representative of ASR.  However, if we are concerned specifically with corruption accuracy or ASR, why should we care about these metrics rather than just using corruption accuracy or ASR directly (or use corruption error/adversarial error if considering training). I think the motivation of studying the correlations between different metrics needs to be made more clear.
- Novelty and significance: The metrics investigated in this paper are all previously proposed metrics so the contribution of this paper is mainly the scope of experiments and analyses presented.  However, I feel like the result that there is no metric that fits all robustness definitions is unsurprising especially since ASR is included as a robustness definition and many works in adversarial robustness have demonstrated tradeoffs between clean accuracy and robustness [1,2].  The significance of the contributions is also a bit unclear to me: what are the main takeaways for researchers or practitioners in this paper?

[1] Zhang, Hongyang, et al. "Theoretically principled trade-off between robustness and accuracy." International conference on machine learning. PMLR, 2019.
[2] Raghunathan, Aditi, et al. "Understanding and Mitigating the Tradeoff between Robustness and Accuracy." International Conference on Machine Learning. PMLR, 2020.

**Questions:**

- Generalization gap and test accuracy: I'm a bit confused by the observed trend where some metrics simultaneously exhibit positive correlation with both generalization gap and test accuracy.  Test accuracy should be inversely correlated with test error which is equal to generalization gap + train error and from the experimental setup it seems like all models are trained until they reach 0.01 cross entropy loss.  If train error is fixed, then generalization gap should be directly correlated with test error which should be inversely correlated with test accuracy.  Do the authors have an understanding of why many of these metrics seem to have this trend of having positive (or negative) correlation with both measures?

---

### Official Review · Reviewer_9CFk · 2024-11-02

**Soundness:** 2
**Presentation:** 2
**Contribution:** 1
**Rating:** 3
**Confidence:** 4

**Summary:**

This paper studies methods for measuring model robustness, proposing four distinct definitions of robustness and introducing various measurement approaches. Using extensive experiments on the Imagenette dataset, it examines the relationships between these measures and the robustness definitions.

**Strengths:**

**Originality**: This paper studies the connections between multiple robustness measures, extending previous work by examining four distinct categories of measures across diverse settings, whereas previous studies typically focus on one or two.

**Quality**: The paper offers a comprehensive set of empirical results on the Imagenette dataset, illustrating the extent to which each measure relates to the proposed robustness definitions.

**Significance**: The findings contribute valuable insights to the community and may lead to new, implicit methods for improving robustness.

**Weaknesses:**

**Ambiguity:** The paper proposes four definitions of robustness. Other than corruption test accuracy and attack success rate, the connections between the other definitions and robustness are unclear.

For instance, the connection between generalization gap and robustness needs further explanation. I suggest the authors provide specific reference to prior work establishing such a connection or clarify the reasoning in the paper. Additionally, the statement ‘A larger Generalization Gap ... indicates poor robustness of a DNN’ (Ln113) requires further justification.

The second robustness definition, clean test accuracy, also raises questions. It’s widely observed that there is a trade-off between clean test accuracy and robustness [A, B], yet the paper claims that higher test accuracy implies greater robustness in a DNN.

This ambiguity represents a significant weakness of the paper.

**Presentation:** The distinction between 'definitions' and 'measures' is unclear. All terms in Sections 3.1 and 3.2 appear to be measurements that can relate to robustness. The primary difference is that measures in 3.1, like corruption test accuracy and attack success rate, more directly reflect model robustness, whereas those in 3.2 represent values that are indirectly connected to robustness. I suggest that the authors clarify the conceptual difference between what they consider 'definitions' versus 'measures' of robustness.

**Evaluation:** This paper is primarily empirical, with two key issues in the evaluation setup.
1. The conclusions are drawn entirely from a single small dataset. This raises questions about the generalizability to larger datasets, such as the full-sized ImageNet. I suggest that the authors discuss the limitations of using only the Imagenette dataset and propose ways to extend the study to larger datasets in future work.
2. The paper relies on FGSM and PGD attacks to assess adversarial robustness. However, evaluating robustness solely with them may overestimate model robustness, as gradient-based methods can fail under certain conditions [C, D]. I suggest the authors to include AutoAttack [D].

[A]: Tsipras et al., Robustness may be at odds with accuracy, ICLR 2019

[B]: Zhang et al., Theoretically principled trade-off between robustness and accuracy, ICML 2019

[C]: Mosbach et al., Logit Pairing Methods Can Fool Gradient-Based Attacks, NeurIPS 2018 Workshop on Security in Machine Learning

[D]: Croce et al., Reliable Evaluation of Adversarial Robustness with an Ensemble of Diverse Parameter-free Attacks, ICML 2020

**Questions:**

The title of this paper is somewhat misleading. It is unclear which sections address the 'improving' aspect.

In the definition of Corruption Test Accuracy (Line 123), the target remains as $t_i$, while in Attack Success Rate (Line 129), the target shifts to $t_i^{adv}$​. This inconsistency requires clarification.

---

### Official Review · Reviewer_czCk · 2024-11-04

**Soundness:** 2
**Presentation:** 2
**Contribution:** 1
**Rating:** 3
**Confidence:** 4

**Summary:**

This paper attempts to compare and unify different measurements of *“robustness”* for deep neural networks and concludes that there are conflicts between these measurements, suggesting that different measurements should be considered for various aspects of robustness. Additionally, the authors study the relationship between the *sharpness* of the loss surface and robustness, arguing that this relationship is largely affected by the batch size.

**Strengths:**

The idea of unifying measurements of robustness for deep neural networks is interesting and aligns with existing research.

**Weaknesses:**

There is a lack of clear definitions for key concepts. I am very confused about the way you define *robustness* in your paper. In Section 3.1, lines 112-115, you regard *robustness* as equivalent to *generalization*. However, I do not believe these terms are interchangeable. *Robustness* has multiple definitions, such as *adversarial robustness* and *natural corruptions*. But it is really rare to define *robustness* as *generalization*. Please refer to the papers by Jiang et al. (2019) and Goodfellow et al. (2014) for clarification on the differences between generalization and adversarial robustness. I suggest providing a formal definition of robustness in the paper instead of a vague reference to various concepts.

This issue becomes more confusing in the following part of Section 3.1, where you define robustness using very different concepts, for example, *Clean Test Accuracy*, *Corruption Test Accuracy*, and *Attack Success Rate*. Rather than treating these as separate concepts, as stated on page 2, lines 99-101, you describe them as different aspects of robustness. And on page 9, lines 480-481, you conclude that there is no single measure representative of robustness across all definitions. However, because these are distinct concepts—related, perhaps, but fundamentally different—they cannot naturally be represented under a single overarching concept. An increase in the generalization gap does not necessarily correspond to an increase in clean test accuracy, especially when training accuracy declines even more.

Some concept definitions are incorrect. On page 3, according to the current definition of Attack Success Rate, ASR would always equal 100%. Please revise this definition. ASR is typically defined as the percentage of instances where an adversarial perturbation successfully alters the model's prediction, if your definition is different, please include a formal definition of your ASR.

**Questions:**

1. On page 4, Section 3.3, you consider both FGSM and PGD as attack methods. I am curious why you did not include more recent and advanced methods, such as Auto-Attack.

2. It is necessary to redefine the concept of *robustness* in your paper. One of your main claims relies on the misuse of this concept, making the conclusion trivial.

3. The content in Section 2 (Related Works) would be more appropriate in the Introduction. I suggest a thorough review of the related works and a more structured presentation of this content.

---

### Note · Authors · 2024-11-15

I have read and agree with the venue's withdrawal policy on behalf of myself and my co-authors.